# Functional Analysis of a Novel *ABL* (*Abnormal Browning Related to Light*) Gene in Mycelial Brown Film Formation of *Lentinula edodes*

**DOI:** 10.3390/jof6040272

**Published:** 2020-11-09

**Authors:** Chang Pyo Hong, Suyun Moon, Seung-il Yoo, Jong-Hyun Noh, Han-Gyu Ko, Hyun A. Kim, Hyeon-Su Ro, Hyunwoo Cho, Jong-Wook Chung, Hwa-Yong Lee, Hojin Ryu

**Affiliations:** 1Department of R&D Planning & Management, Theragen Bio, Suwon 16229, Korea; changpyo.hong@therabio.kr (C.P.H.); seungil.yoo@therabio.kr (S.-i.Y.); royalsiren@naver.com (H.A.K.); 2Department of Biology, Chungbuk National University, Cheongju 28644, Korea; sooym21@gmail.com; 3Forest Mushroom Research Center, National Forestry Cooperative Federation, Yeoju 12653, Korea; korp666@naver.com (J.-H.N.); morchella@hanmail.net (H.-G.K.); 4Division of Applied Life Science and Research Institute of Life Sciences, Gyeongsang National University, Jinju 52828, Korea; rohyeon@gnu.ac.kr; 5Department of Industrial Plant Science and Technology, Chungbuk National University, Cheongju 28644, Korea; hwcho@chungbuk.ac.kr (H.C.); jwchung73@chungbuk.ac.kr (J.-W.C.); 6Department of Forest Science, Chungbuk National University, Cheongju 28644, Korea

**Keywords:** *Lentinula edodes*, mycelium brown film, *abnormal browning related to light*

## Abstract

*Lentinula edodes* is a globally important edible mushroom species that is appreciated for its medicinal properties as well as its nutritional value. During commercial cultivation, a mycelial brown film forms on the surface of the sawdust growth medium at the late vegetative stage. Mycelial film formation is a critical developmental process that contributes to the quantity and quality of the mushroom yield. However, little is known regarding the genetic underpinnings of brown film formation on the surface of mycelial tissue. A novel causal gene associated with the formation of the mycelial brown film, named *ABL* (*Abnormal browning related to light*), was identified in this study. The comparative genetic analysis by dihybrid crosses between normal and abnormal browning film cultivars demonstrated that a single dominant allele was responsible for the abnormal mycelium browning phenotype. Whole-genome sequencing analysis of hybrid isolates revealed five missense single-nucleotide polymorphisms (SNPs) in the *ABL* locus of individuals forming abnormal partial brown films. Additional whole-genome resequencing of a further 16 cultivars showed that three of the five missense SNPs were strongly associated with the abnormal browning phenotype. Overexpression of the dominant *abl-D* allele in a wild-type background conferred the abnormal mycelial browning phenotype upon transformants, with slender hyphae observed as a general defective mycelial growth phenotype. Our methodology will aid the future discovery of candidate genes associated with favorable traits in edible mushrooms. The discovery of a novel gene, *ABL*, associated with mycelial film formation will facilitate marker-associated breeding in *L. edodes.*

## 1. Introduction

*Lentinula edodes*, commonly known as the pyogo or shiitake mushroom, is a white-rot fungus that grows on logs or sawdust by degrading cellulose, hemicellulose, and lignin. *L. edodes*, which belongs to the order Agaricales, is one of the most popular edible mushroom species, accounting for approximately 22% of the global mushroom supply [1]. *L. edodes* is abundant in a range of nutrients, polysaccharides, and unique flavor components, which has led to its use as a source of pharmaceuticals as well as in foods and natural seasonings [2]. Previous research verified the antiviral, antifungal, antioxidant, antitumor, immunomodulatory, and hepatoprotective activities of *L. edodes* and their pharmacological potential [3]. In recent years, demand for *L. edodes* has increased substantially in newer markets in Europe, America, and Africa, as well as in historical markets in Asian countries such as Korea, China, and Japan [4].

*L. edodes* is widely cultivated on sawdust, which accounts for approximately 80% of cultivation media [5]. *L. edodes* cultivation on sawdust involves three main phases. First, the growth substrate is fully colonized by vegetative mycelial growth. Next, brown film formation is induced on the surface of the vegetative mycelium by exposure to light. Finally, in the reproductive phase, initiation of fruiting primordia occurs followed by fruiting body development [6]. The light induction of brown film formation is particularly critical for the development of abundant, high-quality, *L. edodes* fruiting bodies in sawdust cultivation [7]. Our previous study described an abnormal dark yellow-brown film mutant cultivar, Chamaram, and examined its transcriptional regulation [7]. Genome-wide transcriptome analysis of Chamaram revealed that diverse complex signaling and metabolic pathways were involved in the mycelial browning process, including light perception via G protein signaling pathways, downstream melanogenesis, and cell wall degradation [6,7].

The genetic dissection of important agronomic traits in elite cultivars is essential for *L. edodes* breeding programs. Important agronomic traits include those related to yield, quality, bioactive compound production, and disease resilience. Previous studies showed that the most important agronomic traits in *L. edodes*, including rapid mycelium growth, high precocity, favorable fruiting body morphology, and high yield, were quantitative traits controlled by multiple genes or quantitative trait loci (QTLs). These traits were strongly influenced by the environment and exhibited continuous variation [8,9]. Li et al. [9] developed several molecular markers, including InDel and SSR markers, which were associated with several agronomic traits, including a pileus diameter and mass, stipe mass, and single fruiting body mass. To date, improvement of the quantitative or qualitative traits of fruiting bodies has been the main target of *L. edodes* breeding schemes. The brown film formation during the late vegetative stages of *L. edodes* is strongly associated with crop yield and quality, and understanding the genetic basis of the brown film formation is thus important for its breeding programs. It remains unknown whether brown film formation is controlled by several QTLs or by a single gene. The recent publication of whole-genome sequences of *L. edodes* [10,11] will aid the development of high-resolution markers, such as single-nucleotide polymorphisms (SNPs), and facilitate the discovery of novel functional genes, including those associated with brown film formation. The development of markers associated with brown film formation will enhance our understanding of the genetic architecture of the trait and will provide candidate markers for marker-assisted selection (MAS).

In this study, comparative genetic and genomic analyses were used to identify the genetic factors underlying the abnormal mycelial film phenotype in the *L. edodes* Chamaram cultivar. A novel gene of unknown function, *ABL*, was identified and associated with brown film formation, and a single dominant allele of *ABL* was responsible for the abnormal mycelium browning trait. Comparative genomic analysis showed that the dominant allele harbored five missense SNPs in the gene locus. Genome resequencing of 16 *L. edodes* cultivars showed that three of the five SNPs were strongly associated with abnormal mycelial film traits. Overexpression of the dominant *abl-D* allele in a wild-type background conferred the abnormal phenotype on transformants, confirming the role of *ABL* in mycelium browning in *L. edodes*.

## 2. Materials and Methods

### 2.1. Fungal Materials and Cultivation Conditions

*L. edodes* strains were provided by the Forest Service of the National Institute of Forest Science in Korea (http://www.forest.go.kr) and the Forest Mushroom Research Centre (https://www.fmrc.or.kr/) (strains P37-5, Chamaram, Sanjo 701, Sanjo 705, and Sanjo 713). Strains were maintained on potato dextrose agar (PDA) at 25 °C.

To analyze mycelial browning phenotypes, each strain was cultured on 1.2 kg of sterilized oak sawdust medium (4:1 (*v/v*) oak sawdust and rice bran with a water content of approximately 65%). Oak sawdust medium was sterilized at 105 °C for 10 h prior to spawn inoculation. To induce abnormal brown film development, inoculated media were cultured under a 16 h light/8 h dark cycle for 100 days. After 100 days of culture, the mycelial layer on the surface of sawdust media was sampled and used for further analysis.

### 2.2. Comparative Genetic Analysis of Chamaram

To identify genetic components associated with mycelial browning, monokaryons isolated from Chamaram spores were self-mated to form dikaryons. In total, 108 F_2_ progeny strains were analyzed to assess segregation of the normal and abnormal mycelial browning phenotypes. The observed segregation ratio of abnormal browning (AB) and browning (B) was tested for goodness-of-fit to the expected Mendelian ratio using a chi-square test [12]. To assess allele dominance, parental strain P37-5 was cross-mated with monokaryons isolated from spores of Sanmaru 1 (Sanmaru 1-33, Sanmaru 1-36, Sanmaru 1-42) or Sanjo705 (Sanjo 705-12, Sanjo 705-15). The five hybrid dikaryotic strains were cultured on sawdust media and assessed for mycelial browning phenotype.

### 2.3. Genomic Analysis of Mycelial Browning

DNA was extracted from cultured mycelia of monokaryotic strains (P37-5, N27, and N32) and dikaryotic strains (Chamaram and Sanjo713). The monokaryotic P37-5 (P_1_) carried dominant genotype “A” for mycelial browning, whereas N27 (F_2_) and N32 (F_2_) were recessive genotype “a”. Genotypes in the browning of dikaryotic Chamaram (F_1_) and Sanjo713 strains (hybrid of N27 and N32) were heterozygous dominant “A + a” and homozygous recessive “a + a”, respectively (Figure 1c). Mycelia were frozen in liquid nitrogen and ground into powder. DNA extraction was performed using a GenEX Plant Kit (GeneAll, Seoul, Korea) according to the manufacturer’s instructions. Extracted DNA quality was assessed using the 260/280 absorbance ratio and was found to be within an acceptable range (1.8–2.0). DNA sequence libraries were prepared from 1 µg input DNA using a TruSeq Nano DNA Sample Prep Kit according to the manufacturer’s instructions (Illumina, Inc., San Diego, CA, USA). Sheared DNA fragments were subjected to end-repairing, A-tailing, adaptor ligation, and amplification with clean-up. The libraries were subjected to paired-end sequencing with a 100 bp read length using the Illumina HiSeq2500 platform (Illumina). The quality scores of raw reads were assessed with FastQC (v. 0.11.9) (http://www.bioinformatics.babraham.ac.uk/projects/fastqc/). The reads were then processed for quality using Sickle (v. 1.33) (https://github.com/najoshi/sickle) with the following criteria: (i) discard low-quality reads with >10% unknown bases (marked as “N”), and (ii) discard reads with >40% low-quality bases (quality score <20). The genome sequencing dataset is available in the NCBI Sequence Read Archive (SRA) database under the accession number SRR12816459–SRR12816479.

Clean reads were aligned to the draft *L*. *edodes* genome [11] using BWA (v. 0.7.17) [13], and duplicate reads generated by PCR were removed using Picard Tools (v. 1.98) (https://broadinstitute.github.io/picard/index.html). SNP calling was performed using the Genome Analysis ToolKit (GATK) (v. 4.0.5.1) [14] with the following criteria: (i) identification of biallelic SNPs, (ii) minimum read depth ≥10, (iii) minimum genotype quality ≥60, and (iv) no missing alleles among samples. Variants were represented in standard VCF format. Alleles with genotypes common to brown (B) (N27, N32, and Sanjo713 presented as a recessive trait) or abnormal brown (AB) (P37-5 and Chamaram presented as a dominant trait) mycelial films were selected (Figure 2a). Called SNPs were annotated using SnpEff (v. 4.1) [15] with gene predictions as reported by Park et al., 2017 [16].

The full sequence of *ABL* was amplified from genomic DNA extracted from the B17 and P37-5 strains. Sequences, including trait-associated SNPs, were validated by Sanger sequencing (Cosmogenetech, Seoul, Korea). Mycelial tissue for RNA extraction was sampled from each strain, frozen in liquid nitrogen, and ground into powder. RNA extraction was performed using a Hybrid-R™ kit (GeneAll, Seoul, Korea) according to the manufacturer’s instructions. Approximately 2 µg total RNA was reverse transcribed using TOPscript™ RT DryMIX (dT18 plus) (Enzynomics, Daejeon, Korea). Amplification of *ABL* was performed using 1 μL (10 pM) of gene-specific primers (*ABL*-F: 5′ CGGGATCCATGACTTTGGTCCTGTCTCGG 3′, *ABL*-R: 5′ AAGGCCTAACCCTCGTAGTCCTCG 3′), 10 μL of Solg 2 × Taq PCR Smartmix (SolGent Co., Daejeon, Korea), and 1 µL cDNA in a final reaction volume of 20 µL. Thermocycling parameters were as follows: 95 °C for 3 min; 35 cycles of 95 °C for 15 s, 58 °C for 30 s, and 72 °C for 1 min; and a final extension at 72 °C for 5 min. Sequence of *ABL* and *abl-D* are available in the NCBI GenBank under the accession number MW192788 and MW192789, respectively.

### 2.4. Plasmid Construction

The plasmid pCAMBIA1300-gfp, containing EGFP and a NOS terminator between the HindIII and EcoRI sites of pCAMBIA1300, was used for plasmid construction. The *L. edodes* glyceraldehyde-3-phosphate dehydrogenase *(gpd)* promoter [17] was amplified from genomic DNA of strain B17 using primers pLegpd-F (5′-ACGGCCAGTGCCAAGCTTTCGATATCAGTCAGATTGTCA-3′) and pLegpd-R (5′-CGGGATCCGGCCTGAATAGACATGGAAT-3′) or pLegpd-R-BamHI (5′-CGGGATCCGGCCTGAATAGACATGGAAT-3′) for the construction of pLegpd-GFP or pLegpd-*abl-D*-GFP, respectively. The *abl-D* sequence was amplified from strain P37-5 using primers *ABL*-F and *ABL*-inf-R (5′-CGGGATCCGGCCTGAATAGACATGGAAT-3′). The two PCR products were digested with BamHI and ligated, after which the pLegpd-*abl-D* fragment was cloned into pCAMBIA1300-gfp using an EZ-Fusion™ HT cloning kit (Enzynomics, Daejeon, Korea) according to the manufacturer’s protocol.

### 2.5. PEG-Mediated Transformation of Protoplasts

Mycelia for protoplast isolation was cultured in potato dextrose broth (PDB). Two agar plugs were excised from the growing edge of a mycelial colony on a PDA plate, transferred into flasks containing 100 mL of PDB, and incubated at 25 °C for 7–10 days. Mycelia were collected, washed with 0.6 M sucrose, transferred to a 50 mL tube containing 10 mL of enzyme solution (2.5% Lysing Enzymes from *Trichoderma harzianum* (Sigma-Aldrich, St. Louis, MO, USA) dissolved in 0.6 M sucrose), and constantly agitated at 90 rpm for 2 h at 30 °C. Protoplasts were collected by filtering the mixture through Miracloth. Protoplasts were recovered by centrifugation at 3000 rpm for 10 min at 4 °C. Protoplasts were then washed twice in 1 mL of 0.6 M sucrose and recovered by centrifuging at 5000 × *g* rpm for 5 min. Finally, the protoplasts were suspended in 0.6 M sucrose and adjusted to a final concentration of approximately 10^7^ cells/mL.

For fungal transformation, 100 μL of protoplasts were mixed with 20 μg plasmid DNA in a 2 mL tube and incubated on ice for 5 min. Next, 100 μL of PTC solution (40% polyethylene glycol 4000, 10 mM Tris-HCL, pH 7.5, 50 mM CaCl_2_) was added, and the tube was mixed well with gentle flicking. After incubation at room temperature for 5 min, 600 μL of 0.6 M sucrose was added and mixed by inverting. Transformed protoplasts were spread on regeneration plates (PDA containing 0.6 M sucrose and 30 μg/mL of hygromycin B). To detect introduced DNA, specific amplification of the *abl-GFP* sequence was performed using genomic DNA extracted from mycelia and fruit bodies of hygromycin-resistant isolates. To observe GFP fluorescence, transformed cells were pelleted and resuspended in 0.6 M sucrose and incubated for 15 h at 25 °C prior to visualization with a Nikon Eclipse Ti-U (PhotoFluor LM-75 for fluorescence) inverted microscope. Images were captured using a Nikon DS-Ri2 camera with NIS Elements software.

### 2.6. Microstructure of Mycelial Layer

To observe the microstructure of the mycelial layer on the sawdust medium, samples from the mycelial surface were subjected to field emission electron microscopy. Samples were prefixed with 2.5% glutaraldehyde for 2 h and then washed twice with 0.1 M phosphate buffer for 10 min. Samples were postfixed with 0.1% osmium tetroxide for 1 h. After fixation, samples were successively dehydrated with 15%, 30%, 50%, 70%, 80%, 90%, and 95% alcohol, each for 15 min, and then three times with 100% alcohol with isoamylacetate for 10 min. The dehydrated samples were then coated with white gold and observed with field emission scanning electron microscopy (LEO-1530, Carl Zeiss, Oberkochen, Germany).

## 3. Results

### 3.1. Genetic Analysis of Mycelium Browning in L. edodes

During cultivation of *L. edodes*, a brown mycelial film develops on the surface of the sawdust growth substrate after exposure to light as a precursor to mushroom development. Chamaram, a mutant *L. edodes* cultivar, forms an abnormal dark yellow-brown mycelial film when exposed to light at the early vegetative mycelial growth stage [7]. Our previous research [7] demonstrated that the abnormal brown film phenotype in the Chamaram cultivar resulted in defects in fruiting body formation and higher susceptibility to infection with *Trichoderma* spp. (Figure 1a). Electron microscopy of mycelial films showed that hyphal development was impaired in Chamaram compared with a wild-type cultivar, SJ701 (Sanjo701), with normal brown film mycelia (Figure 1b), suggesting that the abnormal phenotype was a heritable trait.

To identify whether the abnormal mycelial film phenotype was genetically determined, the trait was examined in the pedigree of the Chamaram cultivar (Figure 1c). Chamaram is a hybrid line derived from the mating of monokaryotic P37-5 and SJ701-51 strains (Figure 1c; [18]). To test whether the abnormal browning phenotype was determined by homozygotic or heterozygotic alleles, dikaryotic isolates were generated by the self-mating of compatible monokaryotic spores collected from a Chamaram fruiting body, and their mycelial film phenotypes were assessed. In total, 108 progenies were assessed, with 80 isolates exhibiting the abnormal yellow-brown film phenotype and 28 exhibiting the normal brown film phenotype. The phenotypes conformed to a Mendelian 3:1 ratio with a chi-square (χ^2^) value of 0.049 and a *p*-value of 0.842 (Table 1), indicating that a single heterozygotic dominant allele was responsible for the abnormal brown film phenotype. Next, monokaryotic strains Sanmaru 1-33, Sanmaru 1-36, Sanmaru 1-42, Sanjo 705-12, and Sanjo 705-15, which were isolated from spores of the normal browning cultivars Sanmaru 1 and Sanjo 705, were crossed to P37-5, a parental cultivar of Chamaram. By contrast with a wild-type cultivar, SJ701, which displayed a normal mycelial film (Figure 1b), all heterozygotic cross-mated progenies displayed the abnormal mycelium browning phenotype (Figure 1c). These results indicate that the abnormal mycelium browning trait in the Chamaram cultivar is dominant and was inherited from the parental P37-5 monokaryon.

### 3.2. Identification of SNPs Associated with Mycelium Browning in L. edodes

Comparative genomic analysis was used to examine the genetic factors underlying the abnormal yellow-brown mycelial film phenotype. DNA resequencing of two dikaryotic cultivars, Chamaram and SJ713, was performed. SJ713, which exhibited normal mycelial browning, was produced by the self-mating of two monokaryotic isolates, N27 and N32, from Chamaram spores (Figure 1c). Three monokaryotic isolates, P37-5, N27, and N32, were also sequenced. The sequence reads of all five cultivar strains mapped to the reference *L. edodes* genome [11], with average depth and genome coverage of 41.99% and 96.76%, respectively (Appendix A). In total, 776 SNPs were identified in the five strains on the basis of their normal and abnormal mycelium browning phenotypes (Figure 2a; Appendix A).

Most of the 776 SNPs were dispersed throughout the genome; however, eight coding SNPs (612 C > A, 601 A > T, 248 C > G, 247 T > G, 244 G > A, 240 C > G, 40 A > T, and 39 T > G) were found in a single predicted gene, LEG01287 (975 bp in length), located at 2220.8–2221.8 kb of *L. edodes* scaffold 3 (Figure 2b). The predicted LEG01287 gene was validated as a transcribed region using existing RNA-Seq data [11], and expression of the gene was confirmed with a semi-quantitative qRT-PCR (Figure 2c). However, LEG01287 exhibited no similarities to known functional genes, indicative of a novel gene. Five of the eight coding SNPs (612 C > A, 601 A > T, 248 C > G, 240 C > G, and 39 T > G) were predicted to be missense SNPs (Figure 2d). The eight SNPs were validated by Sanger sequencing (Figure 2b and Appendix A), which also identified two additional insertions in the coding region of the Chamaram sequence (Figure 2d). The SNPs were further validated via whole-genome sequencing of 16 *L. edodes* cultivars, 13 of which had normal mycelial browning and three of which had abnormal browning phenotypes. Two of the abnormal cultivars, HG808 and SJ708, exhibited partial browning, and one cultivar, CSI, produced a white mycelial film. Three SNPs (601A > T, 248 C > G, and 247 T > G) were present in heterozygous dominant alleles in the three cultivars, HG808, CSI, and SJ708, with partial browning or white film phenotypes (Figure 2e). These results suggest that three coding SNPs, 601A > T, 248 C > G, and 247 T > G, in LEG01287 were dominantly associated with the formation of abnormal mycelium browning in *L. edodes*. LEG01287 was thus named *Abnormal browning related to light* (*ABL*).

### 3.3. Transformation of L. edodes with the ABL Gene

Functional genetic analysis was used to investigate the biological roles of the newly identified *ABL* gene during mycelial film formation. *L*. *edodes* protoplasts from hyphae of SJ705, a cultivar with a normal mycelium browning phenotype, were transformed with the *abl-D* allele to confirm the association of *abl-D* with the abnormal browning phenotype. Initially, the viability of *L. edodes* protoplasts was assessed, and the regeneration rate was found to be ~2.6%. Cell wall regeneration and cell division started one day after protoplast isolation, and normal hyphal development was observed after three days (Figure 3a). We also confirmed that GFP expression under the control of *glyceraldehyde-3-phosphate dehydrogenase* promoter of *L. edodes* (*pLegpd:GFP*) [17] was stably observed in protoplasts and regenerated hyphae (Figure 3b). The *abl-D* transgene was overexpressed in the protoplasts under the control of the *Legpd* promoter. The *pLegpd:abl-D-GFP* construct was successfully transformed into protoplasts, and the transgene was confirmed in mycelial tissues and fruiting bodies of the transgenic strains by PCR amplification (Figure 3c). A stable GFP signal (Figure 3d) and *abl-D-GFP* mRNA expression (Figure 3e) were confirmed in the transformed hyphae.

### 3.4. Functional Analysis of ABL in Brown Film Formation

To verify the biological roles of *ABL* in mycelium brown film formation, transformants expressing *abl-D-*GFP were crossed with a monokaryotic strain isolated from spores of the normal browning cultivar SJ705. Upon early light exposure, the resulting dikaryotic mycelial tissue harboring *pLegpd:abl-D-GFP* displayed abnormal brown film formation similar to that of Chamaram (Figure 4). The wild-type SJ705 cultivar developed a normal brown mycelial film (Figure 4a). The microstructure of the hyphal layers was examined by using a field emission scanning electron microscope to observe the hyphal surface. Wild-type strains had well-developed mycelial tissues. By contrast, undigested fibers of oak sawdust and shrunken hyphae were observed in the abnormal browning mycelial surfaces of the *pLegpd:abl-D-GFP* transgenic line and the Chamaram cultivar (Figure 4b). These results indicated that the *L*. *edodes* cultivar overexpressing *abl-D-*GFP was unable to form a normal mycelial brown film and was limited in its ability to decompose sawdust. Taken together, these results demonstrate that the newly identified *ABL* gene is associated with mycelial brown film formation in *L. edodes*.

## 4. Discussion

Mycelial brown film formation is characteristic of *L. edodes* sawdust cultivation and is a critical developmental phase, which aids water retention in the sawdust media and enhances the quality and quantity of the fruiting bodies [7]. Abnormal mycelium browning in the Chamaram cultivar increased susceptibility to infection by pathogenic *Trichoderma* spp. and resulted in decreased mushroom yields [7]. In this study, comparative genetic analysis of the dominant abnormal browning trait in Chamaram facilitated the identification of a novel gene, *ABL*, which correlated with mycelial browning phenotypes.

*ABL* is likely to be a genetically driven gene in *L. edodes*. Dihybrid cross-analysis between monokaryotic P37-5 and SJ701-51 lines revealed 20 SNPs in the *ABL* locus (2220.8–2221.8 kb of scaffold 3, SNP positions indicated in Appendix A). The variants were identified as homozygous SNPs in P37-5 (P_1_) and heterozygous SNPs in Chamaram (F_1_), both of which exhibited the abnormal mycelium browning phenotype. Genetic analysis with whole-genome sequencing allowed the identification of *ABL* alleles associated with the formation of partially abnormal brown films. Five of the SNPs were missense SNPs, and they were closely associated with the *ABL* locus in the two abnormal browning cultivars (Figure 2d). Three of the missense SNPs were identified in other abnormal cultivars with white (CSI) or partially browning (HG808 and SJ708) films. These results suggest that these three SNPs are strongly associated with the formation of abnormal mycelial films in *L. edodes*.

The genetic analysis of the *abl-D* allele presents unanswered questions with regard to the developmental regulation of the *L. edodes* life cycle. Several transcriptome-related studies and analytical chemical studies reported that brown film formation in *L. edodes* was closely related to melanin accumulation [6,19]. In addition, our previous studies found that the browning trait was strongly linked to genes involved in mycelial growth and development [7]. Genes involved in the perception of blue light and downstream signaling pathways are known to be involved in mycelial browning and development [7,20]. Furthermore, enzymes related to carbohydrate metabolism are critical for the regulation of mycelial developmental processes [7]. The *ABL* gene identified in this study is of unknown function, and its effects on mycelium browning could not be associated with any specific signaling or metabolic processes. However, our data indicate that three important SNPs are likely to be linked as dominant traits for the disruption of mycelial development, resulting in an abnormal partial browning phenotype (Figure 2 and Figure 4). Integrative functional genomic analysis of the links between *ABL* and the regulation of melanin accumulation and mycelial development will clarify the roles of *ABL* in mycelial brown film formation and reproductive fruit body development.

Genetic transformation is widely used for the investigation of gene function in model organisms. *Agrobacterium* spp. are commonly used as vectors for the generation of transgenic plants, yeasts, and filamentous fungi [21,22]. *Agrobacterium tumefaciens*-mediated transformation has been used in diverse mushroom species such as *Agaricus bisporus* [23,24,25], *Flammulina velutipes* [26,27], *Hebeloma cylindrosporum* [28], *Hypsizygus marmoreus* [29], *L*. *edodes* [30], *Morchella importuna* [31], *Pleurotus eryngii* [32], *Pleurotus ostreatus* [33], and *Volvariella volvacea* [34]. Furthermore, T-DNA-mediated genetic disruption for screening genes related to mycelial development was reported in *Ophiocordyceps sinensis* [35]. *Agrobacterium tumefaciens*-mediated transformation can be used with protoplasts, spores, hyphae, and mushroom tissues [36]. However, this method can result in mixtures of untransformed and transformed cells within transformants when multicellular mycelial tissues are used [37]. In this study, an improved polyethylene glycol (PEG)-mediated protoplast transformation method was used for functional analysis of *ABL* in *L. edodes*. Protoplast transformation has some disadvantages, such as the requirement for high concentrations of viable protoplasts, low transformation efficiencies, high percentages of transient transformants, and frequent integration of transforming DNA into multiple loci [38]. However, protoplast transformation is simpler than *Agrobacterium tumefaciens*-mediated methods, and homozygotic mycelial tissues can be reliably recovered (Figure 3 and Figure 4). Our PEG-mediated transformation methodology could be widely applicable for the transformation of filamentous fungi for functional genetic analysis.

Our aim in this study was to identify *L. edodes* genes associated with mycelial browning upon exposure to light in the early stage of sawdust cultivation. The results of this study will facilitate the development of molecular markers for *L. edodes* horticultural breeding programs. The integrative approach used in this study involves comparative genetic and genomic analyses and is a powerful research methodology that supports the discovery of new genes and helps to elucidate phenotypic correlations with particular genes under a range of environmental conditions. In addition, this research strategy could be used as a standard method for identifying trait-related genes in edible mushrooms and will improve mushroom breeding by contributing to MAS.

## Figures and Tables

**Figure 1 jof-06-00272-f001:**
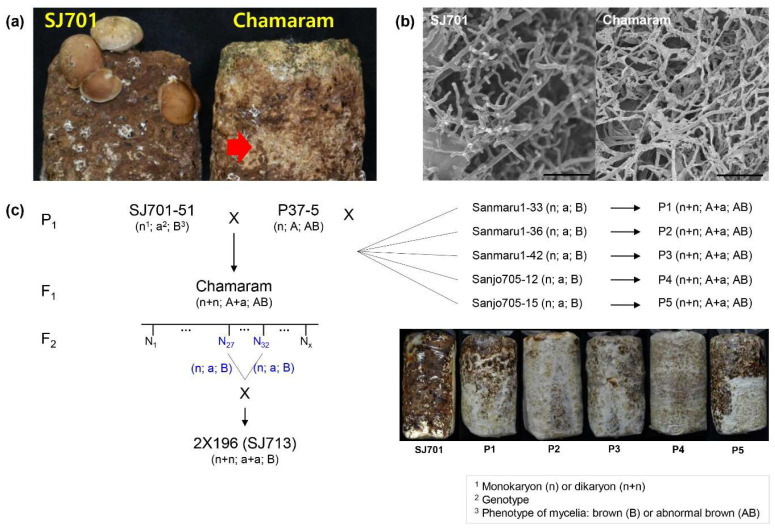
Genetic analysis of abnormal brown film formation in *L. edodes* cultivar, Chamaram. (**a**) Surface of sawdust cultivation media colonized with Chamaram and Sanjo701 (SJ701) cultivars after light exposure at the early vegetative mycelial growth stage. The abnormal brown mycelial film formation is indicated by the arrow. (**b**) Microstructure of SJ701 and Chamaram mycelial films. Scale bar = 20 μm. (**c**) Genetic analysis of abnormal brown film formation in the Chamaram cultivar using dihybrid lines. The karyotype ^1^, hypothetical genotype ^2^ related to mycelia browning pattern, phenotypes ^3^ of mycelia (B: normal brown film, AB: abnormal partial brown film) were represented. The putative dominant monokaryotic P37-5 line was crossed with the five indicated recessive monokaryotic strains. The brown film phenotypes of the heterozygotic dikaryon strains (P1–P5) are shown (right).

**Figure 2 jof-06-00272-f002:**
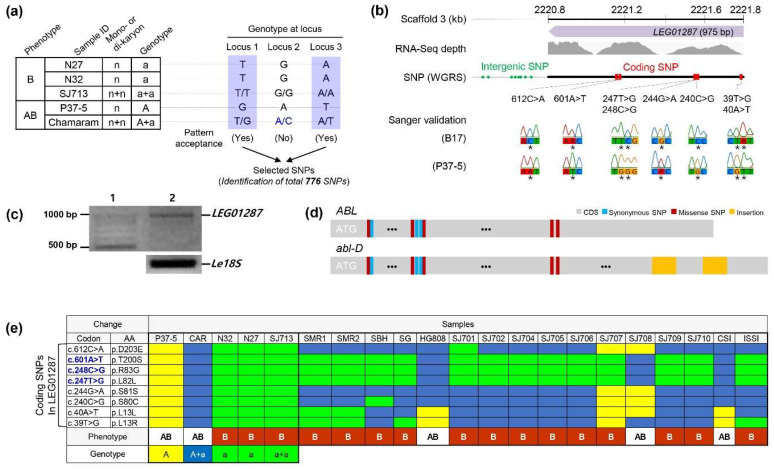
Identification of a dominant *abl*-*D* allele associated with abnormal brown film formation. (**a**) Identification of single-nucleotide polymorphisms (SNPs) through comparison of genome sequences from selected strains. A total of 776 SNPs that the allele combination of genotypes “A” and “a” at each loci met the phenotype pattern (A or AB) were identified. The purple-color shaded boxes indicate an allele that is accepted as a valid SNP. (**b**) Schematic structure of the candidate gene responsible for abnormal brown film formation with the locations of predicted SNPs indicated. The symbol * means SNP position. (**c**) RT-PCR analysis of LEG01287 gene expression. *Le18 S* was used as an internal control. Lane 1—1 Kb molecular weight marker; Lane 2—*Le18 S* and *LEG01287* PCR products. (**d**) Schematic diagram of *ABL* (wild-type, upper) and *abl*-*D* (mutant type, lower). Polymorphism locations are indicated. The orange-color boxes indicate insertions of the *abl-D* allele. (**e**) Identification of the most relevant trait-associated SNPs. CAR, Chamaram; SJ713, Sanjo713; SMR1, Sanmaru1; SMR2, Sanmaru2; SBH, Sanbaekhyang; SG, Songgo; SJ701, Sanjo701; SJ702, Sanjo702; SJ704, Sanjo704; SJ705, Sanjo705; SJ706, Sanjo706; SJ707, Sanjo707; SJ708, Sanjo708; SJ709, Sanjo709; SJ710, Sanjo710; CSI, Chamsongi; and ISSI, Iseulsongi. Polymorphisms of *ABL* are represented by yellow for dominant genotype “A” (i.e., P37-5); blue for heterozygous dominant genotype “A + a’” (i.e., CAR); green for recessive genotypes “a” (i.e., N32 and N27) or “a + a” (i.e., SJ713), identical to B17 (reference) alleles. Mycelial film phenotypes are indicated as brown (B) and abnormal brown (AB).

**Figure 3 jof-06-00272-f003:**
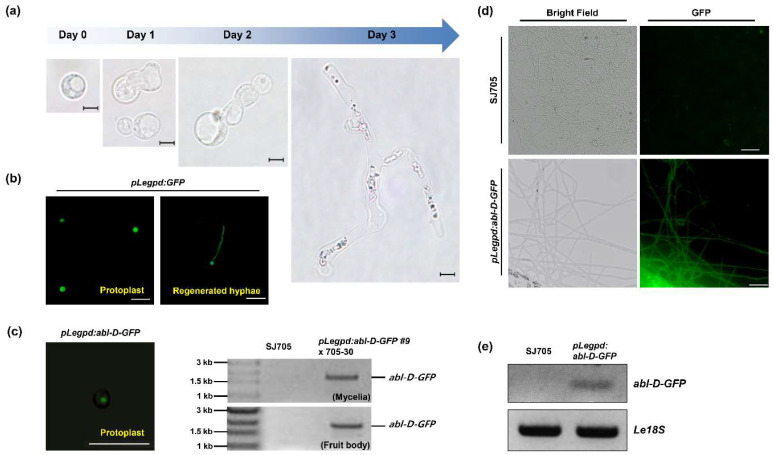
Generation of monokaryotic strains overexpressing *abl*-*D*-GFP by protoplast transformation (**a**) Regeneration of *L*. *edodes* protoplasts. Scale bar = 10 μm. (**b**) Stable expression of GFP under the control of the native *Legpd* promoter. GFP fluorescence was observed at 15 h (Protolast) and 3 days (Regenerated hyphae) after transformation. Scale bar = 50 μm. (**c**–**e**) Expression analysis of the *abl*-*D*-GFP transgene. GFP fluorescence was observed in protoplasts ((**c**), left) and regenerated hyphae (**d**). Scale bar = 50 μm. PCR amplification of the transgene from the genomic DNA of transgenic mycelia and fruit bodies ((**c**,) right), and from cDNA derived from the mRNA of transgenic mycelia (**e**).

**Figure 4 jof-06-00272-f004:**
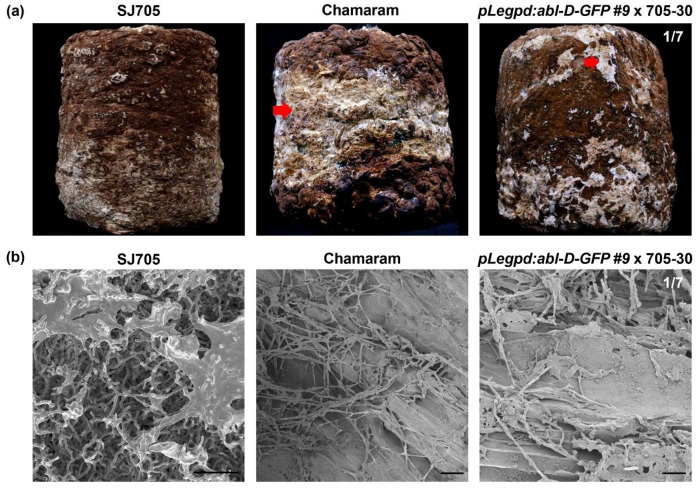
Mycelial brown film phenotype in transgenic strains overexpressing *abl*-*D*-GFP. (**a**) Mycelial brown film formation on sawdust media in the transgenic line, wild-type SJ705, and Chamaram. Images were taken after cultivation in conditions conducive to abnormal browning induction. The abnormal brown mycelial film is indicated by the arrow. (**b**) Scanning electron microscopy of mycelia from SJ705, Chamaram, and the *pLegpd:abl-D-GFP* dikaryotic transgenic line. Scale bar = 20 μm.

**Table 1 jof-06-00272-t001:** Chi-square test for Mendelian segregation of abnormal brown film formation in F_2_ generation.

Generation	Abnormal	Normal	Total	Expected Ratio	χ^2^	*p*-Value
F2	80	28	108	3:1	0.049	0.824

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
