# Peer review of "Functional Analysis of a Novel *ABL* (*Abnormal Browning Related to Light*) Gene in Mycelial Brown Film Formation of *Lentinula edodes"

_jof, 2020, doi:10.3390/jof6040272_

Round 1
Reviewer 1 Report
In their manuscript, Hong et al. describe the identification of a gene responsible for abnormal browning in the edible mushroom Lentinula edodes using genetic and genomic analyses. In addition, they establish a method for protoplast transformation of L. edodes. The results are of interest to researchers investigating fungal development or the cultivation of edible mushrooms. There are just few points where the manuscript should be improved:
- The genotypes A and a are mentioned in lines 116-117, but not explained. Similarly, in Figure 2a, it is not clear why the genotype combination of Locus 2 was not selected as SNP, because the requirement of homokaryotic and heterokaryotic strains with phenotypes B or AB would be met by the allele combinations of A and a as well in Locus 2 as in Locus 1 and Locus 3, unless there is prior knowledge about which genotype would be the dominant one.
- Figure 2a: What does the blue shading for Locus 1 mean?
- Figure 2e: The color scheme of the Figure does not fit with the legend. According to the legend, there should be an orange color for the alternative allele (is this the blue color in the actual figure?). Also, is the phenotype "PB" in the one that is labelled "AB" in the legend and text? Also, the Figure shows strain P37-5 with genotype AA, but in Figure 1, this strain is described as a monokaryon with genotype A.
- Line 362 should read Morchella (instead of Mochella)
Author Response
Comment 1: In their manuscript, Hong et al. describe the identification of a gene responsible for abnormal browning in the edible mushroom Lentinula edodes using genetic and genomic analyses. In addition, they establish a method for protoplast transformation of L. edodes. The results are of interest to researchers investigating fungal development or the cultivation of edible mushrooms. There are just few points where the manuscript should be improved:
Response: We thank Reviewer 1 for positive and critical comments on our manuscript. We have responded to the comments by Reviewer 1 as follows.
Comment 2: The genotypes A and a are mentioned in lines 116-117, but not explained. Similarly, in Figure 2a, it is not clear why the genotype combination of Locus 2 was not selected as SNP, because the requirement of homokaryotic and heterokaryotic strains with phenotypes B or AB would be met by the allele combinations of A and a as well in Locus 2 as in Locus 1 and Locus 3, unless there is prior knowledge about which genotype would be the dominant one.
Response: Thank you for offering your valuable, constructive advice. For the revision, we first clarified the genotypes of five samples, that were used for genetic analysis through dihybrid crosses and were resequenced, by considering the composition of nuclei within a cell in the formation of monkaryon and dikaryon; ‘A’ or ‘a’ for monokaryon (n), ‘A+a’ (represented as heterozygous genotype) for hetero-dikaryon (n+n), and ‘a+a’ (represented as homozygous genotype) for homo-dikaryon (n+n). According to this definition, we have revised the Method section and Figure 2a. In the Method section, the details of genotypes of the five samples has been added as follows; DNA was extracted from cultured mycelia of monokaryotic strains (P37-5, N27, and N32) and dikaryotic strains (Chamaram and Sanjo713). The monokaryotic P37-5 (P1) carried dominant genotype ‘A’ for mycelial browning, whereas N27 (F2) and N32 (F2) were recessive genotype ‘a’. Genotypes in the browning of dikaryotic Chamaram (F1) and Sanjo713 strains (hybrid of N27 and N32) were heterozygous dominant ‘A+a’ and homozygous recessive ‘a+a’, respectively (Figure 1c) (yellow color-highlighted at Line 116 ~ 120).
We have newly modified Figure 2a illustrating how to select SNPs from five samples used for genetic analysis through dihybrid crosses. The information of phenotype of mycelia [normal browning (B) and abnormal browning (AB)], karyotypes (mono- (n) or di- (n+n) karyons), and genotypes of those five samples were added (left panel in Figure 2a). A total of 776 SNPs that the allele combination of genotypes ‘A’ and ‘a’ at each of loci met the phenotype pattern (A or AB) were identified (SNPs indicating ‘Yes’ in the right panel of Figure 2a). We have also revised the corresponding Figure legend (Line 274 – 277).
Comment 3: Figure 2a: What does the blue shading for Locus 1 mean?
Response: Sorry for incomplete illustration. We have newly revised Figure 2a. In Figure 2a, the purple shading part indicates a valid allele that was accepted if the allele combination of genotypes ‘A’ and ‘a’ at each locus meets the phenotype pattern (A or AB).
Comment 4: Figure 2e: The color scheme of the Figure does not fit with the legend. According to the legend, there should be an orange color for the alternative allele (is this the blue color in the actual figure?). Also, is the phenotype "PB" in the one that is labelled "AB" in the legend and text? Also, the Figure shows strain P37-5 with genotype AA, but in Figure 1, this strain is described as a monokaryon with genotype A.
Response: We apologize to inaccurate description and confusing the review of manuscript. We also thank to the good advice for the labeling of phenotype. First, ‘orange color’ in the Figure 2e legend was from our mistake. ‘yellow’ is correct. Besides this, the descriptions on genotypes and colors shown in Figure 2e have been revised as follows; Polymorphisms of ABL are represented by yellow (‘not orange-colored’) for dominant genotype ‘A’ (i.e. P37-5); blue for heterozygous dominant genotype ‘A+a’ (i.e. CAR); green for recessive genotypes ‘a’ (i.e. N32 and N27) or ‘a+a’ (i.e. SJ713), identical to B17 (reference) alleles. Mycelial film phenotypes are indicated as brown (B) and abnormal brown (AB) (Line 285 ~ 288). We also changed phenotype ‘PB’ to ‘AB’ at the bottom of Figure 2e. In addition, genotypes were changed as follows; ‘AA’ to ‘A’ for P37-5, ‘Aa’ to ‘A+a’ for CAR, ‘aa’ to ‘a’ for N32 and N27, and ‘aa’ to ‘a+a’ for SJ713.
Additionally, we found to omit the description of orange-color box indicating ‘insertion’ in abl-D in Figure 2d legend. We have revised it (Line 280 ~ 281).
Comment 5: Line 362 should read Morchella (instead of Mochella)
Response: Sorry for our mistake. We revised this error (Line 370)

Reviewer 2 Report
I read the paper entitled “Functional analysis of a novel ABL (Abnormal browning related to light) gene in mycelial brown film formation of Lentinula edodes”, submitted to Journal of Fungi. This paper shows a key gene associated with the formation of mycelial brown film, ABL, from genetic analysis and SNPs identification. The knowledges on light induced brown film formation are important for develop of sawdust-based cultivation of Lentinula edodes. The data seem to be solid and are well presented in the manuscript. My minor comments are shown below.
L247
The authors should indicate GenBank Accession number of LEG01287 (GAW08769?)
L272 Figure 2
There is no “orange” in Figure 2(e).
L318-332
This paragraph in the Discussion is similar to the expressions in the Introduction. The authors should avoid repetitive expressions and correct this part.
Author Response
Comment 1: I read the paper entitled “Functional analysis of a novel ABL (Abnormal browning related to light) gene in mycelial brown film formation of Lentinula edodes”, submitted to Journal of Fungi. This paper shows a key gene associated with the formation of mycelial brown film, ABL, from genetic analysis and SNPs identification. The knowledges on light induced brown film formation are important for develop of sawdust-based cultivation of Lentinula edodes. The data seem to be solid and are well presented in the manuscript. My minor comments are shown below.
Response: We thank Reviewer 2 for positive, helpful comments on our manuscript. We have responded to the comments by Reviewer 2 as follows.
Comment 2: L247
The authors should indicate GenBank Accession number of LEG01287 (GAW08769?)
Response: According to the comment of Reviewer 2, we submitted sequences of ABL and abl-D to NCBI GenBank and have received two accession numbers MW192788 for ABL and MW192789 for abl-D. These accession numbers were added in the revised manuscript. (Line 154 – 156)
Comment 3: L272 Figure 2
There is no “orange” in Figure 2(e).
Response: Sorry for our mistake in Figure 2 legend. ‘Orange color’ in the Figure 2e legend was from our mistake. ‘Yellow’ is correct. We have revised the error with additional descriptions on genotypes and colors shown in Figure 2e (Line 285 ~ 288).
Comment 4: L318-332
This paragraph in the Discussion is similar to the expressions in the Introduction. The authors should avoid repetitive expressions and correct this part.
Response: We deeply appreciate this critical comment. We carefully amended the first paragraph of discussion part (Line 334 ~ 340)